# *C. cochlearium* 2316 Ameliorates High-Fat Diet-Induced Obesity and Metabolic Syndrome Risk Factors via Enhanced Energy Expenditure and Glucose Homeostasis

**DOI:** 10.3390/nu17243848

**Published:** 2025-12-10

**Authors:** Wenjun Zhu, Paba Edirisuriya, Qing Ai, Fei Yang, Jiangqi Tang, Kai Nie, Xiangming Ji, Samira Soltanieh, Maesha Musarrat, Md Abdul Alim, Zerui Liao, Kequan Zhou

**Affiliations:** 1Department of Nutrition and Food Science, Wayne State University, Detroit, MI 48202, USA; 2Department of Nutritional Sciences, The Pennsylvania State University, University Park, PA 16802, USA

**Keywords:** *C. cochlearium*, dietary supplementation, body weight gain, glucose homeostasis, high-fat diet, energy expenditure, gut microbiome

## Abstract

Objectives: This study investigated the potential beneficial effects of a probiotic candidate, *Clostridium cochlearium* 2316, in modulating physiological and metabolic markers in mice with high-fat diet-induced obesity (DIO). Methods: C57BL/6 DIO mice were assigned to three groups (ad libitum): standard low-fat control (LF, 10% fat), high-fat diet (HF, 60% fat), and high-fat diet supplemented with approximately one billion CFU/day of CC2316 via daily oral gavage for 16 weeks. Results: After 16 weeks, the CC group exhibited 17.3% lower body weight gain (*p* < 0.001) and significant fat mass decrease (*p* < 0.0001) compared to HF mice. Serum biochemistry showed that CC2316 supplementation resulted in a 27.7% reduction in fasting blood glucose (*p* < 0.05), a 58.4% reduction in fasting insulin (*p* < 0.01), and an 89.4% improvement in HOMA-IR score (*p* < 0.05). Furthermore, serum total cholesterol level decreased dramatically by 40.2% in the CC group (*p* < 0.001). Despite a higher caloric absorption rate (*p* < 0.001), CC mice demonstrated a significant beneficial shift in energy expenditure, characterized by an increased basal metabolic rate (*p* < 0.05), higher energy expenditure (*p* < 0.05), and an elevated respiratory quotient (RER) (*p* < 0.05), alongside increased physical activity (*p* < 0.05). Conclusions: This investigation strongly suggests that CC2316 supplementation mitigates the adverse effects of HFD-induced obesity by modulating whole-body energy metabolism, positioning it as a potential aid to lower risk factors associated with metabolic syndrome. The precise mechanisms linking the gut microbiome to altered energy substrate utilization are discussed and suggested for further investigation.

## 1. Introduction

Obesity is a major risk factor for many severe diseases related to metabolic syndrome. With comorbidities such as type 2 diabetes (T2D), life expectancy and quality dramatically decrease without early prevention and/or treatment [1,2]. High-fat diets (HFDs) are a primary driver of this pathology, promoting excessive weight gain and chronic low-grade inflammation that collectively lead to the development of metabolic syndrome. Effective preventative or therapeutic strategies are urgently needed to mitigate high-fat diet-induced effects. Intervention research has been targeting various systemic conditions and related underlying diseases. One type of interventions is to initiate intense change through lifestyle modification such as dietary improvement and increasing physical activity [3]. In recent years, increasing effort has been implemented to target the gut microbiome, which is clearly associated with energy metabolism and disease status due to dysbiosis. Dysbiosis is an imbalance in the gut microbial community induced by HFD consumption, and it has been correlated with increased intestinal permeability, adiposity, and insulin resistance [4,5]. Thus, understanding the gut–body relationship in terms of microbes and organ-system interaction has become a new approach to the investigation of disease manifestation and potential prevention or treatment [6]. For example, microbial diversity creates a complex network targeting the metabolism of various small and large molecules from the digestion process. Throughout the digestive tract, subsequent nutrient products, as well as toxins and other potentially harmful substances, are further being utilized and metabolized by gut microorganisms. Due to such intricate properties of the gut microflora, as well as due to the underlying mechanism associated with metabolic diseases and other pathological conditions such as IBD and colon cancer, it is crucial to understand and discover potential preventative strategies [7]. Therefore, modulating the gut microbial profile through the introduction of beneficial microorganisms, known as probiotics, represents a promising and non-invasive avenue for managing metabolic disease [8].

According to previous findings, there are a number of indications that several metabolites are strongly associated with obesity. Of particular interest is glutamate, a neurotransmitter that has been linked to obesity onset and prevalence [9,10]. A high level of glutamate has been shown in mice with high-fat diet-induced obesity [11]. The key connection lies in the role of the microbial fermentation of glutamate in the production of short-chain fatty acids (SCFAs) such as acetate and butyrate.

Several clostridium species are known to carry out such reactions through glutamate mutase activities [12,13]. Within this genus, *Clostridium cochlearium*, a common and non-pathogenic inhabitant of the mammalian gut flora (rat, mouse, and human) [14,15,16], was our research interest. Our laboratory utilized a specific isolate, *C. cochlearium* strain 2316 (CC2316), and our published work has established that its supplementation significantly reduces body weight gain and exerts favorable metabolic benefits in animal models of diet-induced obesity [17,18]. Given the accumulating evidence suggesting that strain-specific properties dictate the efficacy of probiotics, continuous validation and elucidation of mechanisms are essential for translating these findings.

Building upon these established findings, the primary objective of the current investigation was twofold: first, to examine the probiotic consistency and reproducibility of the beneficial metabolic effects of CC2316 supplementation in C57BL/6 mice, and second, to explore the potential mechanisms underlying its anti-obesity effects. We specifically hypothesized that CC2316 intervention would mitigate adiposity and improve key metabolic markers (e.g., glucose, insulin, and cholesterol) by favorably altering host energy metabolism.

## 2. Materials and Methods

### 2.1. Bacterial Preparation

The probiotic strain utilized in this investigation was *Clostridium cochlearium* CC2316, an isolate from our institutional nutrition and food science laboratory. The initial identification of the strain was confirmed through a combination of phenotypic characterization and molecular analysis. Specifically, species identity was verified using 16S rRNA gene sequencing and comparative culture against a commercially available reference strain of *C. cochlearium* purchased from ATCC (Manassas, VA, USA). For animal administration, the CC2316 strain was cultured under strict anaerobic conditions using specific media and methods recommended for *Clostridium* species [19]. Upon the end of the growth phase, culture solution was collected and centrifuged to yield bacterial pellet, which was subsequently washed with sterile phosphate-buffered saline and mixed with 25% glycerol in medium to reach a final concentration of 10^10^ CFU/mL. Viability was assessed using serial dilution and plating methods to reach desired viable concentration for subsequent experiment. Samples were flash-frozen and stored under −20 °C and used within one week of each preparation. During experiment, bacterial samples were prepared daily at 30 min–1 h prior to oral gavage using thawed *C. cochlearium* pellet and were then resuspended in sterile water.

### 2.2. Mouse Study

The experimental protocol was approved by the Institutional Animal Care and Use Committee (IACUC) of Wayne State University for the current study. Cohorts of 36 6–8-week-old male C57BL/6 mice (Charles River Laboratories, Wilmington, MA, USA) were housed under a 12 h day/night cycle, with room temperature kept at 24 °C ± 1 °C and with moisture at 40% ± 10%. The mice’s diets were a high-fat diet (D12492M) containing 5.24 kcal per gram with 60% of calories from fat and 20% of calories from carbohydrates and a low-fat diet (D12450J) containing 3.85 kcal per gram with 10% from fat and 70% from carbohydrates; both were purchased from Research Diets Inc. (New Brunswick, NJ, USA). Details of purified diets are shown in Table 1. After 1 week of acclimatization with the low-fat control diet, mice were randomly assigned into three groups (*n* = 12, 6 mice per cage). The high-fat diet (HF) control and low-fat diet (LF) control groups were fed with 100 µL of sterile water vis oral gavage; food and water were given ad libitum. The experimental group (CC) was fed with 100 µL of *C. cochlearium* culture containing approximately 10^9^ CFU via oral gavage and provided with the high-fat diet.

The experimental duration was set to 16 weeks; food intake and body weight were monitored weekly; fasting blood glucose was measured every four weeks. At week 16, body composition was measured for fat and lean mass using an EchoMRI-100 analyzer (EchoMRI, Houston, TX, USA). Upon termination, all mice were euthanatized by exposure to CO_2_ and subsequent cervical dislocation. Blood collection was performed to extract serum, which was flash-frozen in liquid nitrogen for subsequent analyses of fasting insulin and other serum metabolites. Intestinal content and tissues (liver, kidney, and fat) were collected, weighed, and quenched immediately using liquid nitrogen. Serum and tissue samples were stored at −80 °C until analysis. Partial liver tissues were sectioned and fixed in 10% (*v*/*v*) paraformaldehyde/PBS solution. Then, samples were embedded in paraffin wax for slicing and subsequent staining with hematoxylin and eosin (H&E). Final slides were photographed at 100 µm resolution using Nikin Eclipse digital microscope for comparison.

### 2.3. Glucose Homeostasis and Serum Lipid Assessment

An oral glucose tolerance test (OGTT) was performed in week 15. After an 8 h food deprivation with only water given, all mice were administrated glucose solution (10% *w*/*v* in sterile water) at a dosage of 1 g/kg of body weight via oral gavage. Blood glucose was measured at t = 0, 15, 30, 60, and 120 min with an Accu-check glucometer (Roche, Indianapolis, IN, USA). Incremental areas under the curve (AUCs) of glucose response were calculated using standard trapezoid method.

Fasting insulin was measured at the end of the study (week 16) from collected serum using the ultra-sensitive mouse insulin ELISA kit (Crystal Chem, Doners Grove, IL, USA) [20], and homeostatic model assessment of insulin resistance (HOMA-IR) was calculated as follows: fasting insulin (mU/L) × fasting glucose level (mg/dL)/405. The fasting glucose values used were from the last measurement prior to euthanasia and serum collection.

Serum cholesterol and triglyceride assessment were performed using Cholesterol and Triglycerides liquid reagents kits (Pointe Scientific, Canton, MI, USA). Sample serum used was from the end of week 16 post fasting process.

### 2.4. Metabolic Study

At week 15, each mouse from HF and CC was transferred and caged individually for five days (with 2-day acclimatization) in a TSE PhenoMaster metabolic chamber system (TSE systems, Chesterfield, MO, USA) for the purpose of measuring the mice’s individual energy expenditure and activity level. The housing environment was kept the same as the previous group housing conditions, with the same light (day)/dark (night) cycle. Total food consumption, weight changes, and activity levels (by movement distance) were measured, along with respiratory parameters including heat emission and the respiratory exchange ratio (RER) via CO_2_/O_2_ production. The above indirect calorimetry results were separately analyzed based on day vs. night periods for corresponding metabolic status. In addition, total food consumption from each cage was measured to determine individual average daily intake. Fecal samples were also collected to determine and analyze caloric output using a Bomb Calorimeter (Parr, Moline, IL, USA).

### 2.5. Inflammatory Markers and Safety Evaluation

Serum inflammatory markers and biomarkers for hepatotoxicity were measured using ELISA assay kits according to the manufacturer’s instructions, which included an Alanine transaminase (ALT) assay kit (BioAssay Systems, Hayward, CA, USA) [21], aspartate transaminase (AST) assay kit (BioAssay Systems, Hayward, CA, USA) [22], and a γ-Glutamyltransferase (GGT) assay kit (Sigma-Aldrich, St. Louis, MO, USA) [23]. In addition, kidney toxicity from urea concentration was also measured and converted into blood urea nitrogen (BUN) concentration using a QuantiChrom urea assay kit (BioAssay Systems, Hayward, CA, USA) [24].

### 2.6. Statistical Analysis

All measurements were analyzed and presented as means ± SD. Data normality was tested using the Shapiro–Wilk test. Statistical analysis was performed using Student’s *t*-test, one-way ANOVA by Dunnett’s test, and two-way ANOVA by the Sidak post hoc test for multiple comparisons. Results were considered statistically significant at *p* < 0.05. Significance was indicated with * for *p* < 0.05, ** for *p* < 0.01, *** for *p* < 0.001, and **** for *p* < 0.0001; ns stood for non-significance. Spearman correlations were used to calculate species correlations for various parameters, setting parameters at a coefficient >0.5 or <−0.5 and *p* < 0.05. Statistical analyses were performed with GraphPad Prism (V 8.00, La Jolla, CA, USA).

## 3. Results

### 3.1. Reduced Body Weight Gain and Modified Body Composition

The high-fat diet induced significant body weight gain in both the *C. cochlearium* CC2316 (CC) group and the high-fat diet (HF) control group compared to the low-fat control group. However, mice with CC2316 supplementation presented a significantly lower rate of body mass increase than the HF group; significance was first observed from the third week of treatment and progressively grew until the end of week 16 (Table 2, Figure 1a). At the end of the dietary treatment, the HF group showed a total weight gain of 101.37 ± 10.82%, compared to 85.57 ± 17.84% in the CC group, representing a significant 17.3% reduction in weight gain in the CC-treated mice (*p* < 0.0001, Figure 1b).

Body composition analysis demonstrated that the weight reduction in the CC group was primarily driven by changes in fat mass deposition (Figure 2a,b). The HF group gained an average of 5.52 g more fat mass than the CC group. Consequently, the HF group showed a significantly higher percentage of fat mass and a lower percentage of lean mass, resulting in a substantially elevated fat/lean ratio compared to the CC group. Tissue analysis further revealed that liver (*p* < 0.001) and kidney weight (*p* < 0.0001) were significantly increased in HF mice compared to CC mice. Macroscopic and microscopic examination of liver tissue clearly showed significant fatty cell deposits, and an overall fatty liver appearance was induced in HF mice, while the tissue condition was noticeably healthier in the CC group (Figure 2c,d). No significant differences were observed in colon weight or length across the groups.

### 3.2. Improved Blood Glucose Homeostasis

Serum analysis confirmed that CC2316 supplementation significantly improved glucose homeostasis markers. At the end of the 16-week intervention, the CC group maintained a normal range of fasting blood glucose levels at 175.18 ± 34.05 mg/dL, which was comparable to the LF control group on a low-fat diet (Figure 3a) and significantly lower than the HF group (223.75 ± 50.16 mg/dL, *p* < 0.05, Table 3). The oral glucose tolerance test (OGTT) showed a significantly elevated glucose response in the HF group, beginning at the 15 min mark and persisting throughout the 120 min testing period. The area under the curve (AUC) for glucose concentration was 60.20% lower in the CC group compared to the HF group (*p* < 0.0001, Figure 3b,c). Furthermore, plasma fasting insulin levels were significantly elevated in the HF group (192.36 ± 36.43 mU/L) compared to the CC group (121.46 ± 46.17 mU/L, *p* < 0.01). This translated to a HOMA-IR score of 104.80 ± 34.79 in HF mice, indicating severe insulin resistance, while the CC mice’s score (55.32 ± 29.72, *p* < 0.05) suggested partial alleviation.

### 3.3. Improved Lipid Markers

Regarding lipid markers, serum total triglyceride concentration was not significantly different between the groups. However, total cholesterol was significantly lower in CC mice than HF mice (*p* < 0.001, Table 3).

### 3.4. Inflammatory Marker and Organ Toxicity Measurement

Inflammatory and organ (liver and kidney) toxicity markers including ALT, AST, GGT, and BUN showed no significant elevation in the CC group compared to the LF group, suggesting a protective or neutral effect of the CC intervention.

### 3.5. Shifted Energy Balance and Metabolic Processes

Food intake (total energy intake) did not differ significantly across all groups. However, the subsequent analysis of feed efficiency (daily total caloric intake per weight gain) yielded a significantly higher ratio in the HF group (2.03 ± 0.17%) compared to the CC group (1.68 ± 0.32%, *p* < 0.01). When considering energy absorption by accounting for fecal energy output, the total caloric absorption was significantly lower in the CC group (9.60 ± 0.09 kcal/day/mouse) compared to the HF group (10.09 ± 0.04 kcal/day/mouse, *p* < 0.001).

Indirect calorimetry analysis revealed significant shifts in energy expenditure and substrate utilization. The respiratory exchange ratio (RER) was significantly higher in the CC group compared to the HF group (*p* < 0.05). Furthermore, both energy expenditure (EE) per mouse body weight during the light cycle (*p* < 0.05) and the dark cycle (*p* < 0.05) were significantly elevated in the CC group (Table 4).

Basal metabolic rate (BMR), calculated based on the average heat emission from the ten lowest consecutive time points (kcal per 24 h) and divided by individual mouse body weight, was significantly higher in the CC group (234.85 ± 22.28 kcal/day/kg) compared to the HF group (212.20 ± 20.15 kcal/day/kg, *p* < 0.05). In addition to basal energy shift, mice in the CC group also showed a higher daily physical activity level, measured as distance traveled (meters per day) compared to the HF group (Table 4).

## 4. Discussion

This investigation consistently demonstrated that CC2316 supplementation mitigates the adverse effects of a high-fat diet (HFD) in C57BL/6 mice, confirming our hypothesis and establishing the probiotic consistency of CC2316 compared to our previous work [17,18]. The effects of a high-fat diet on C57BL/6 mice included body weight gain, glucose intolerance, hyperglycemia, increased adipose tissue, and increased liver weight, as expected [25]. Dietary supplementation with CC2316 ameliorated these adverse effects and reduced several markers associated with obesity risk factors, which demonstrates consistency with our previous intervention [17]. In terms of appearance and body weight, the biometric measurement showed clear differences and leaner shapes in the CC group over the duration of the study, which was further confirmed via body composition analysis showing that the reduction in total weight gain observed in the CC group was accompanied by a significant decrease in fat mass (5.85% decrease) and an increase in lean mass (2.64% increase). This shift underscores the anti-obesity activity of the CC treatment and its positive influence on body composition. Furthermore, there was a significant reduction in liver and kidney weight in the CC group—61.0% (*p* < 0.0001) and 19.6% (*p* < 0.01), respectively—compared to HF mice, suggesting that CC2316 treatment may prevent HFD-induced ectopic fat deposition and organ hypertrophy. The correlation of phenotypic data from liver weight against body weight and fat percentage indicated clear and significant positive correlations (Figure 4a,b). Although detailed kidney analysis was not our object of interest and showed no noticeable relation to body weight, there could be a potential sequela of obesity and metabolic syndrome involving the endocrine system [26].

The analysis of serum biomarkers strongly supports the hypothesis that CC2316 alleviates metabolic dysfunction. The significant improvements in fasting glucose, fasting insulin, and the HOMA-IR index in the CC group confirm that the supplementation substantially reduces HFD-induced insulin resistance and impaired glucose homeostasis. These findings align with research suggesting a strong correlation between hepatic steatosis (indicated by increased liver weight) and insulin resistance (Figure 4c,d) [27,28].

While serum triglyceride levels were not significantly altered—potentially due to the pre-blood collection fasting procedure—the significant reduction in total cholesterol in the CC group suggests a benefit of the CC treatment in promoting downstream fatty acid or cholesterol oxidation. Given the varied outcomes reported in the literature regarding HFDs and serum cholesterol [29,30], future studies must investigate more detailed parameters involving cholesterol metabolism, including transport signaling pathways, de novo synthesis, and enterohepatic bile acid circulation, to fully elucidate the mechanism.

When looking at obesity pathogenesis, there seems to be a simple causality from a positive energy balance, where continuous lower expenditure contributes to adiposity. Conventionally, this has been the target of approaches for lowering obesity risk by reducing intake and/or increasing expenditure [31]. Since the total energy intake was not different among the groups in our investigation regardless of high-fat diet or low-fat diet, this suggests that CC2316 supplementation reduced body weight gain by increasing energy expenditure. Combining fecal energy output and calorie absorption, feed efficiency analysis confirmed a greater reduction in energy storage as adipose tissue in the CC group, which was shown in body composition data. Furthermore, mouse energy expenditure measured through indirect calorimetry analysis revealed greater heat emission during the 12 h light cycle from the HF mice compared to the CC mice (Table 4. Day EE, *p* < 0.05) but not during the dark cycle (Night EE). These were further analyzed due to their significant association with individual body weight (Figure 4e), with a coefficient of 0.611 for daytime and 0.602 for nighttime energy expenditure (*p* < 0.001). Based on the weights, it was confirmed that the CC mice exhibited significantly higher total energy expenditure during both the light (121.28 ± 12.22 kcal/day/kg) and dark (130.70 ± 14.13 kcal/day/kg) cycles compared to the HF group (light: 108.01 ± 11.02 kcal/day/kg; dark: 108.01 ± 11.02 kcal/day/kg). Although the mechanisms between elevated energy expenditure and influences of gut microbial shift are yet unclear, studies have suggested probiotics could alter microflora-associated host gut barrier functions in terms of nutrient absorption, inflammatory response, and other signaling pathways involving intestinal metabolites [32,33].

Further analysis of mouse BMR suggests that the CC group not only had an overall increase in total energy expenditure but also a significantly elevated basal metabolism. Due to the strong association with body mass, as mentioned previously, we used one of the three main methods for an accurate evaluation of BMR based on individual weight change in mice [34]. Since both BMR and EE were significantly negatively associated with fat mass (Figure 4f,g), this elevated energy output is the likely mechanism explaining the lower body weight gain despite the HFD challenge.

Interestingly, we observed a significant positive correlation between BMR and RER (r = 0.729, *p* < 0.0001) (Figure 4h), where the CC group generally showed higher ratios than the HF group, suggesting a relatively greater reliance on carbohydrate metabolism. This finding presents a potential mechanistic paradox: while weight reduction through fat mass decrease typically correlates with increased fatty acid oxidation and a lower RER, the higher RER observed in the CC group may be associated with an increased production of microbial short-chain fatty acids (SCFAs) via the glutamate fermentation pathway, as hypothesized in the Introduction [35]. SCFAs are rapidly oxidized, contributing to overall energy expenditure and potentially shifting the RER. Accordingly, subsequent research should prioritize in vivo SCFA production and organ-specific substrate utilization to elucidate this metabolic profile.

Beyond SCFA production, the significant reduction in cholesterol suggests that the anti-obesity effect may be linked to the interaction between gut microbiota and host bile acid metabolism [36,37]. Bile acids have important signaling and regulatory roles in cholesterol and fat synthesis/oxidation. Probiotic-induced alterations in the intestinal environment and metabolite transportation could stimulate cholesterol re-uptake through peripheral adipose tissue [38]. Therefore, we suggest additional experimentation involving signaling pathways for energy metabolism and some of the intestinal epithelial transport proteins such as Farnesoid X receptor (FXR), G-protein-coupled bile acid receptor (TGR5), and organ-specific AMP-activated protein kinase [33,39,40]. Combined with ongoing sequencing and metabolomic analysis of CC mice’s gut content, these approaches will provide a comprehensive understanding of the mechanism network behind CC2316 [41,42,43].

The safety evaluation on CC2316 supplementation showed no adverse effects in the current mouse study, which is consistent with prior reports [17,18,44]. Based on the mice’s general appearance, daily food intake status, fecal matter consistency, and activity levels, there was no indication of abnormality or significant difference compared to LF control group throughout the 16-week experimentation. Serum biomarkers of glucose homeostasis and lipid profile were further confirmed to show a protective capability of the treatment from high-fat induced dysfunctions. In addition, the liver and kidney toxicity panel (ALT, AST, GGT, and BUN) indicated that there was no significant difference in test results under CC2316 supplementation when compared to the LF group. Moreover, the CC group showed lowered responses in those parameters compared to the HF group, suggesting a long-term protective potential against an increasing adverse effect from high-fat diet. The dosage used for the animal study was 10^9^ CFU/day/mice, which is equivalent to 33.3 billion CFU/kg using an average mouse weight of 30 g. For a human equivalent dosage for an average of 70 kg adult, this corresponds to 10^11.3^ CFU based on our calculations [45]. Considering the overall evaluation of the toxicity and serum data, this dosage is potentially safe for long-term use, which we also intend to study as an experiment objective in the future.

## 5. Conclusions

In conclusion, this investigation confirms that CC2316 supplementation effectively mitigates high-fat diet-induced obesity and metabolic syndrome risk factors in C57BL/6 mice. The primary mechanism of action involves a significant beneficial modulation of energy metabolism, characterized by an increased basal metabolic rate and overall energy expenditure. Further multi-omics and signaling pathway analyses are warranted to fully elucidate the complex network linking this unique bacterium to host physiology.

## Figures and Tables

**Figure 1 nutrients-17-03848-f001:**
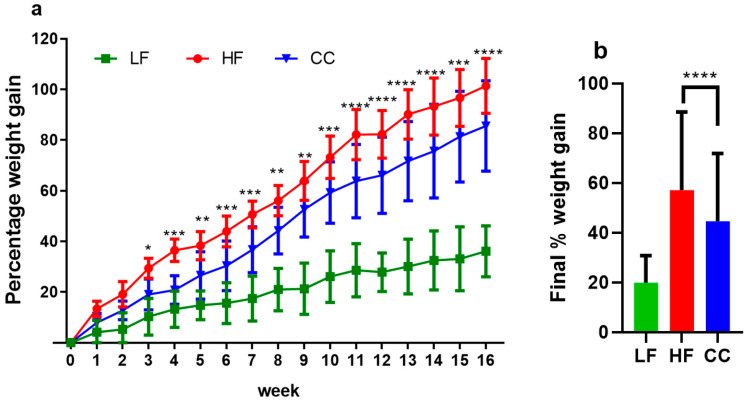
(**a**) Average percentage body weight changes over 16-week period. (**b**) Final percentage weight gained at week 16 compared to initial weight. Values depicted are means ± SD, *n* = 12, and * *p* < 0.05, ** *p* < 0.01, *** *p* < 0.001, and **** *p* < 0.0001 indicate significant differences between the HF and CC groups for every week’s value.

**Figure 2 nutrients-17-03848-f002:**
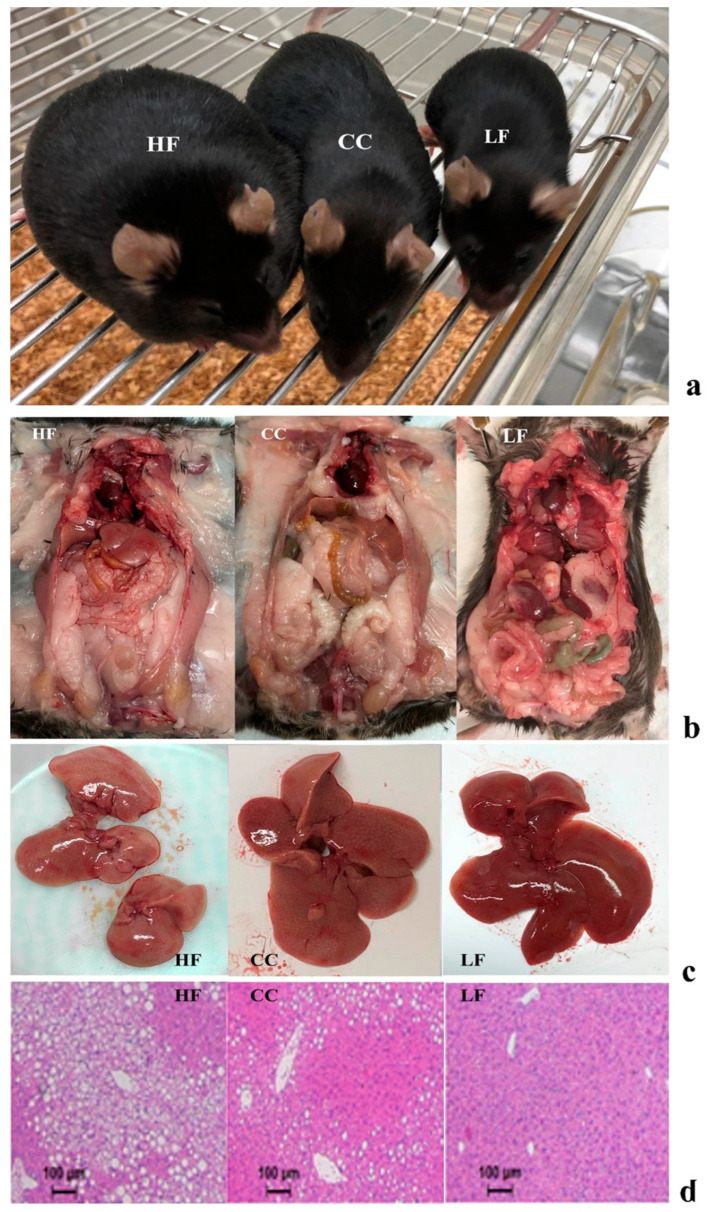
(**a**) Side-by-side comparison of mouse appearance at end of experiment. (**b**) Comparison of dissected mouse body cavities to show size, organs, and fat tissue deposition. (**c**) Liver tissue comparison showing texture and appearance. Images depicted are placed from left to right as HF vs. CC vs. LF mouse. (**d**) H&E staining of liver tissue at 100 µm resolution.

**Figure 3 nutrients-17-03848-f003:**
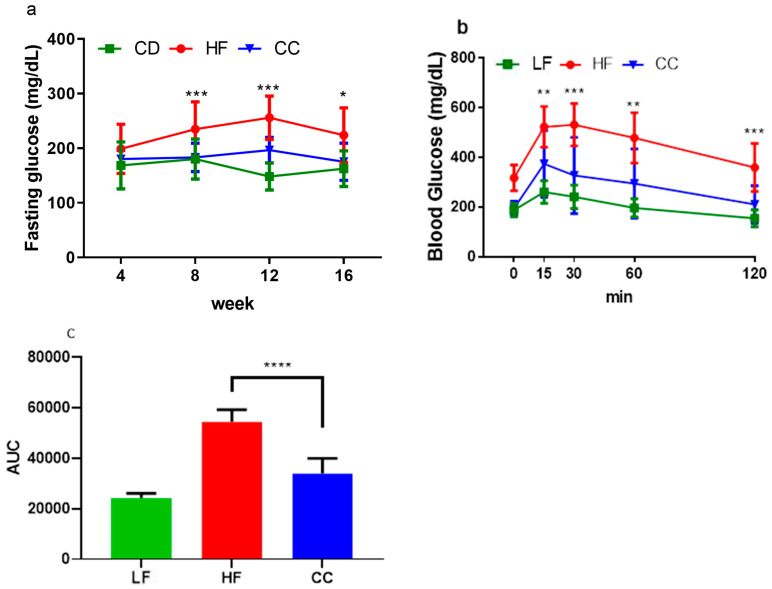
(**a**) Average fasting blood glucose tested every 4-week period. (**b**) Average blood glucose measurement for oral glucose tolerance test over 120 min period. (**c**) Average AUC calculated from OGTT levels from (**b**), where baseline was set from 0. Values depicted are means ± SD, *n* = 12; * *p* < 0.05, ** *p* < 0.01, *** *p* < 0.001, and **** *p* < 0.0001 indicate significance between HF and CC groups.

**Figure 4 nutrients-17-03848-f004:**
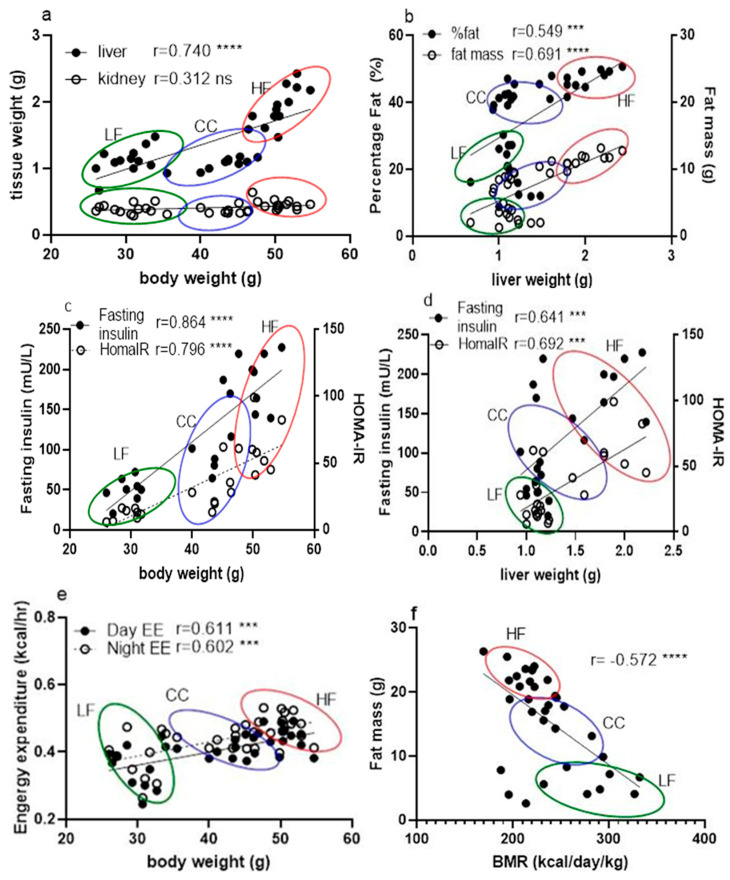
Correlation between phenotypic data across LF, HF, and CC groups analyzed together using linear regression for calculation of Pearson coefficient (*n* = 35). (**a**) Correlation between tissue mass (liver and kidney) vs. mouse body weight measurement. (**b**) Correlation between fat percentage and mouse liver weight. (**c**) Correlation comparison of fasting insulin vs. body weight and HOMA-IR vs. body weight. (**d**) Correlation between fasting insulin level and liver weight. (**e**) Correlation between hourly daytime energy expenditure and hourly nighttime energy expenditure against mouse body weight. (**f**) Correlation between BMR and mouse individual fat mass. (**g**) Correlation between BMR and feed efficiency of individual mice. (**h**) Correlation between BMR and RER from individual mice. Data points are grouped and noted with individual r values shown on each figure along with significance notation: *** *p* < 0.001, and **** *p* < 0.0001.

**Table 1 nutrients-17-03848-t001:** Nutrient composition and caloric content of low-fat diet (LF) and high-fat diet (HF) used in experiment.

	LF (D12450J)		HF (D12492M)	
Ingredient	gm	kcal	gm	kcal
Casein, 30 Mesh	200	800	200	800
L-Cystine	3	12	3	12
Corn Starch	506.2	2024.8	0	0
Maltodextrin 10	125	500	125	500
Sucrose	68.8	275.2	68.8	275
Cellulose, BW200	50	0	50	0
Soybean Oil	25	225	25	225
Lard	20	180	245	2205
Mineral Mix S10026	10	0	10	0
DiCalcium Phosphate	13	0	13	0
Calcium Carbonate	5.5	0	5.5	0
Potassium Citrate, 1 H_2_O	16.5	0	16.5	0
Vitamin Mix V10001	10	40	10	40
Choline Bitartrate	2	0	2	0
**Overall**	**gm%**	**kcal%**	**gm%**	**kcal%**
Protein	19.2	20	26	20
Carbohydrate	67.3	70	26	20
Fat	4.3	10	35	60
Total		100		100
kcal/gm	3.85		5.24	

**Table 2 nutrients-17-03848-t002:** Progressive weight changes and terminal mouse body composition and tissue weight.

	Group			*p* Value
	LF	HF	CC	HF vs. CC
Body weight, g	30.53 ± 2.76	51.22 ± 2.36	43.67 ± 3.96	***
Percentage weight gain, %	36.09 ± 10.02	101.37 ± 10.82	85.57 ± 17.84	****
Fat mass, g	5.74 ± 2022	22.86 ± 1.99	17.34 ± 2.28	****
Percentage fat, %	19.39 ± 7.42	47.10 ± 2.62	41.25 ± 2.71	**
Percentage lean, %	19.98 ± 3.17	21.19 ± 1.40	19.16 ± 1.45	*
Fat/lean ratio	66.99 ± 7.38	43.72 ± 2.56	46.36 ± 2.82	ns
Liver, g	0.30 ± 0.14	1.08 ± 0.12	0.91 ± 0.11	***
Kidney, g	1.12 ± 0.19	1.90 ± 0.35	1.18 ± 0.26	****
Colon weight, g	0.39 ± 0.06	0.46 ± 0.07	0.37 ± 0.05	**
Colon length, cm	0.11 ± 0.02	0.12 ± 0.02	0.10 ± 0.02	0.051

Values depicted are means ± SD, *n* = 12, and * *p* < 0.05, ** *p* < 0.01, *** *p* < 0.001, and **** *p* < 0.0001 indicate significant differences between the HF and CC groups for every week’s value.

**Table 3 nutrients-17-03848-t003:** Blood biochemistry from terminal serum collection.

	Group			*p* Value
	LF	HF	CC	HF vs. CC
Fasting Glucose, mg/dL	162.58 ± 32.7	223.83 ± 50.16	175.18 ± 34.05	*
Fasting Insulin, mU/L	49.51 ± 15.59	192.36 ± 36.43	121.46 ± 46.17	**
HOMA-IR	19.42 ± 6.84	104.80 ± 34.79	55.32 ± 29.72	*
Cholesterol, mg/dL	83.97 ± 18.79	158.18 ± 26.23	112.84 ± 20.12	***
Triglyceride, mg/dL	96.39 ± 12.25	81.43 ± 16.03	78.97 ± 14.73	ns
				**LF vs. CC**
ALT, U/L	3.78 ± 2.46	23.54 ± 6.88	9.30 ± 4.45	ns
AST, U/L	4.90 ± 1.42	9.61 ± 5.57	5.46 ± 4.04	ns
GGT, U/L	1.70 ± 0.29	1.67 ± 0.87	1.63 ± 0.28	ns
BUN, mg/dL	19.77 ± 2.07	26.29 ± 5.30	20.40 ± 1.35	ns

HOMA-IR calculated using fasting insulin (mU/L) × fasting glucose (mg/dL)/405, *n* = 8–10. * *p* < 0.05, ** *p* < 0.01, and *** *p* < 0.001 indicate significant differences between the HF and CC groups for every week’s value.

**Table 4 nutrients-17-03848-t004:** Energy balance.

	Group			*p* Value
	LF	HF	CC	HF vs. CC
Food intake, kcal/day/mice	10.42 ± 0.74	11.35 ± 1.25	10.66 ± 1.30	ns
Fecal energy, kcal/day/mice	0.79 ± 0.005	1.29 ± 0.09	1.12 ± 0.04	****
Calorie absorption, kcal/day/mouse	9.352 ± 0.005	9.60 ± 0.09	10.09 ± 0.04	***
Feed efficiency, %	0.67 ± 0.21	2.03 ± 0.17	1.68 ± 0.32	**
RER	0.882 ± 0.061	0.773 ± 0.013	0.789 ± 0.018	*
BMR, kcal/day/kg	264.41 ± 50.85	212.20 ± 20.15	234.85 ± 22.28	*
Distance traveled, m/day	2219.17 ± 1824.09	1103.76 ± 469.81	2186.68 ± 1256.12	*
Day EE, kcal/h	0.356 ± 0.064	0.445 ± 0.037	0.414 ± 0.031	*
Night EE, kcal/h	0.383 ± 0.068	0.476 ± 0.043	0.446 ± 0.035	ns
Day EE/wt, kcal/day/kg	146.37 ± 29.21	108.01 ± 11.02	121.28 ± 12.22	*
Night EE/wt, kcal/day/kg	158.25 ± 28.76	115.58 ± 12.47	130.70 ± 14.13	*

RER: respiratory exchange ratio; BMR: basal metabolic rate; Day EE/wt; daytime energy expenditure divided by weight; Night EE: nighttime energy expenditure. Day and nighttime measurement was made during light and dark cycle of 12 h period, respectively; *n* = 12. * *p* < 0.05, ** *p* < 0.01, *** *p* < 0.001, and **** *p* < 0.0001 indicate significant differences between the HF and CC groups for every week’s value.

## Data Availability

The original contributions presented in this study are included in the article. Further inquiries can be directed to the corresponding author.

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
