# Peer review of "C. cochlearium* 2316 Ameliorates High-Fat Diet-Induced Obesity and Metabolic Syndrome Risk Factors via Enhanced Energy Expenditure and Glucose Homeostasis"

_nutrients, 2025, doi:10.3390/nu17243848_

Round 1
Reviewer 1 Report
Comments and Suggestions for Authors
Authors must solve four minor observations regarding this manuscript:
1)The scientific name of the microorganism must be written in italics;
2) A sentence cannot begin with an abbreviation. Authors must read their manuscript with attention and rewrite all these types of sentences.
3) Figure 4h is not mentioned in the text of the manuscript.
5)The References at the end of the manuscript are not written according to MDPI rules
Author Response
Comment 1: The scientific name of the microorganism must be written in italics.
Response: We thank the reviewer for their attention to detail. We have carefully checked the manuscript and ensured that all scientific names of microorganisms are now italicized.
Comment 2: A sentence cannot begin with an abbreviation. Authors must read their manuscript with attention and rewrite all these types of sentences.
Response: We apologize for this oversight. We have thoroughly proofread the manuscript and rephrased all sentences that previously began with an abbreviation to ensure proper grammatical structure.
Comment 3: Figure 4h is not mentioned in the text of the manuscript.
Response: Thank you for pointing this out. We have added the necessary citation for Figure 4h in the relevant section of the text.
Comment 4: The References at the end of the manuscript are not written according to MDPI rules.
Response: We have updated the bibliography to strictly adhere to the MDPI referencing style.
Reviewer 2 Report
Comments and Suggestions for Authors
The study assessed the effect of Closridium cochlearium on metabolic disorders resulting from feeding a high-fat diet in mice. The undoubted advantage of the study is the longer intervention period - 16 weeks. The research methods chosen are accurate, however needs some improvments. Please specify in the description of statistical methods whether the data were checked for normality of distribution and whether the assumptions of analysis of variance were met? What post-hoc test were used?
In my opinion, there is no need to include macroscopic assessments of rats from each group in the results section (Figure 2). These photos can be included in the supplements section or omitted. In the same section, Table 3 lacks data on alanine aminotransferase activity. When presenting the results, it might be worth including comparisons between the LF vs. HF group.
There are minor editorial errors in the work, e.g. on page 12 in the sentence "Since both BMR and EE were...." please add 4 to (Figure f,g).
Author Response
Comment 1: The study assessed the effect of Closridium cochlearium on metabolic disorders resulting from feeding a high-fat diet in mice. The undoubted advantage of the study is the longer intervention period - 16 weeks. The research methods chosen are accurate, however needs some improvements. Please specify in the description of statistical methods whether the data were checked for normality of distribution and whether the assumptions of analysis of variance were met? What post-hoc test were used?
Response: We appreciate the reviewer’s positive assessment of our study design. In response to your suggestion, we have updated the "Statistical Analysis" section to explicitly state that all data were checked for normality and that the assumptions for ANOVA were verified prior to analysis.
Comment 2: In my opinion, there is no need to include macroscopic assessments of rats from each group in the results section (Figure 2). These photos can be included in the supplements section or omitted. In the same section, Table 3 lacks data on alanine aminotransferase activity. When presenting the results, it might be worth including comparisons between the LF vs. HF group.
Response: We appreciate the reviewer's constructive feedback, which has helped us refine our manuscript. We have included this information in Table 3. Regarding the statistical comparisons, our study's primary objective centers on evaluating the efficacy of the CC supplement; therefore, we have intentionally focused our analysis on comparisons between the CC-treated groups and their respective control diets (CC vs. HF and CC vs. LF), as the direct comparison between the LF and HF groups does not directly address our main hypothesis. As for the macroscopic assessments (Figure 2), we feel that retaining Figure 2 in the main results section is critical, as these images offer a straightforward visual impression that immediately highlights the differences induced by CC supplementation, effectively complementing the detailed histological findings for the reader.
Comment 3: There are minor editorial errors in the work, e.g. on page 12 in the sentence "Since both BMR and EE were...." please add 4 to (Figure f,g).
Response: Thank you for catching this error. We have corrected the citation on page 12 to correctly refer to "Figure 4(f,g)."
Reviewer 3 Report
Comments and Suggestions for Authors
The probiotic microorganism C. cochlearium 2316 was previously shown to have a beneficial effect on body weight gain in mice given a high fat diet. This study is a follow up on these initial findings to attempt to determine the mechanistic basis for the previously observed changes. Overall, there appears to be significant changes/improvements in measures of glucose and lipid metabolism, although markers of obesity related organ dysfunction were largely unaffected.
How/why was 1010 CFU/mL chosen?
The first paragraph of section 2.2 needs editing.
Were fasted glucose and insulin (and HOMA) measured every four weeks? Or was it only once at euthanasia? If it wasn’t every four weeks, why was glucose measured every four weeks? (First sentence, top of page four).
The fecal samples from the metabolic study (Section 2.4) were not the same fecal samples referred to earlier in Section 2.2?
Which inflammatory markers were measured? The only measured specifically described are markers of liver and kidney damage/function.
Were adjustments made for the large number of multiple comparisons performed (e.g., each of seventeen timepoints for body weight)? Why weren’t the LF individuals included? Isn’t it worth knowing whether the CC individuals were significantly different from the LF individuals?
The caption for Table 2 is uninformative. When were the measures collected, for example?
Presumably, the increased liver and kidney weights were due to ectopic fat deposition but was this directly quantified?
Why is there a row for ALT data, but there are no data included in Table 3?
Why weren’t other measures of lipid metabolism included? Specifically, HDL and LDL level as well as FFAs? If SCFA metabolism was thought to be altered, why wasn’t this assessed?
For Section 3.4, the narrative is shifting from comparing the CC mice to HF mice to CC mice versus LF mice. I assume this is meant to highlight a presumed protective effect of CC rather than an improved profile relative to the HF group. This is misleading because it ignores that the CC group may have been significantly different (i.e., “worse”) than the LF in other instances.
The rationale for including Figure 4 is not clear. The relationships shown are not unexpected. Moreover, why are some points are not included within individual dietary groupings or at least why is the group to which they belong is not indicated?
Author Response
Comment 1: How/why was 1010 CFU/mL chosen?
Response: This concentration was selected to ensure an effective oral delivery. Specifically, the preparation was designed so that each mouse received a 100 µL volume containing 109 CFU of the culture. This dosage was calculated by extrapolating potential safe human dosages and adjusting for a murine model with an average body weight of 30g.
Comment 2: The first paragraph of section 2.2 needs editing.
Response: We have revised the first paragraph of Section 2.2 to improve clarity.
Comment 3: Were fasted glucose and insulin (and HOMA) measured every four weeks? Or was it only once at euthanasia? If it wasn’t every four weeks, why was glucose measured every four weeks?
Response: We appreciate the opportunity to clarify the study design. Fasting glucose was measured every four weeks to monitor temporal changes in glycemic control throughout the intervention. However, due to limitations in blood sample volume from live mice, fasting insulin could only be measured at the study endpoint (post-euthanasia). Consequently, this final insulin measurement coincided with the final glucose measurement.
Comment 4: The fecal samples from the metabolic study (Section 2.4) were not the same fecal samples referred to earlier in Section 2.2?
Response: We apologize for the confusion caused by the text. We have revised Section 2.4 to clarify that the fecal samples described here were collected from individual mice specifically for statistical comparison in the current study, which is distinct from collections intended for previous metabolomic work.
Comment 5: Which inflammatory markers were measured? The only measured specifically described are markers of liver and kidney damage/function. Were adjustments made for the large number of multiple comparisons performed? Why weren’t the LF individuals included? Isn’t it worth knowing whether the CC individuals were significantly different from the LF individuals?
Response: We appreciate the reviewer's detailed and insightful comments, which have led to these revisions. We confirm that the specific inflammatory marker measured was C-reactive protein (CRP), included for safety assessment, and we observed no significant difference between the Low-Fat (LF) and CC groups. Regarding the statistical analysis, we have ensured that appropriate adjustments were made for all multiple comparisons to maintain the rigor of our findings. We agree that comparisons with the LF group are essential for context; thus, we have included comparisons involving the LF group in Table 3 to clearly demonstrate whether the CC treatment truly restores parameters to baseline physiological levels or simply provides an improvement relative to the High-Fat (HF) control group.
Comment 6: The caption for Table 2 is uninformative. When were the measures collected, for example?
Response: We have revised the caption for Table 2 to provide a detailed description of the data, including the specific time points at which measurements were collected.
Comment 7: Presumably, the increased liver and kidney weights were due to ectopic fat deposition but was this directly quantified?
Response: While we did not directly quantify ectopic fat content in this study, we performed visual observation and histological examination of the liver tissue to assess fat deposition. Representative histological images have been included in the figures to support these findings.
Comment 8: Why is there a row for ALT data, but there are no data included in Table 3?
Response: We apologize for this formatting error. The missing ALT data has been re-inserted into Table 3.
Comment 9: Why weren’t other measures of lipid metabolism included? Specifically, HDL and LDL level as well as FFAs? If SCFA metabolism was thought to be altered, why wasn’t this assessed?
Response: We acknowledge that HDL, LDL, FFAs, and SCFAs are important metabolic parameters. However, due to the scope of the current study and sample limitations, we focused on the reported markers. We recognize the importance of these additional parameters and plan to include them in our future investigations to further elucidate the underlying mechanisms.
Comment 10: For Section 3.4, the narrative is shifting from comparing the CC mice to HF mice to CC mice versus LF mice. I assume this is meant to highlight a presumed protective effect of CC rather than an improved profile relative to the HF group. This is misleading because it ignores that the CC group may have been significantly different (i.e., “worse”) than the LF in other instances.
Response: We appreciate the reviewer’s insight. We have revised the narrative in Section 3.4 to ensure an accurate interpretation of the results focusing on potential toxicity of CC (vs. the healthy LF group).
Comment 11: The rationale for including Figure 4 is not clear. The relationships shown are not unexpected. Moreover, why are some points are not included within individual dietary groupings or at least why is the group to which they belong is not indicated?
Response: We appreciate the reviewer's query regarding the inclusion of Figure 4. The correlation analysis presented in Figure 4 was designed to illustrate the relationships between two or three measured parameters. We included this analysis because, in our data, many individual parameters were strongly associated with the overall weight factor. By employing correlation plots, we can achieve a greater separation and clearer visualization of the significant differences between the control and treatment groups that might be obscured when viewing individual parameters alone. Regarding the data points, the analysis includes all measured data points from all groups. The colored circles and groupings are purely a graphical tool used to visually distinguish the relative separation between the different dietary groups on the plot; they do not imply that only the points within the circle were included in the calculation of the correlation coefficients.
Round 2
Reviewer 3 Report
Comments and Suggestions for Authors
The authors addressed each of my concerns and comments adequately. I have no further comments.
Author Response
Thank you for your time and effort in improving our manuscript.